# A Stop-Gain Mutation within *MLPH* Is Responsible for the Lilac Dilution Observed in Jacob Sheep

**DOI:** 10.3390/genes11060618

**Published:** 2020-06-04

**Authors:** Christian J. Posbergh, Elizabeth A. Staiger, Heather J. Huson

**Affiliations:** 1Department of Animal Science, Cornell University, Ithaca, NY 14853, USA; eas0115@auburn.edu; 2Department of Animal Sciences, Auburn University, Auburn, AL 36849, USA

**Keywords:** *Ovis aries*, coat color, whole-genome sequencing, genomics

## Abstract

A coat color dilution, called lilac, was observed within the Jacob sheep breed. This dilution results in sheep appearing gray, where black would normally occur. Pedigree analysis suggested an autosomal recessive inheritance. Whole-genome sequencing of a dilute case, a known carrier, and sixteen non-dilute sheep was used to identify the molecular variant responsible for the coat color change. Through investigation of the genes *MLPH*, *MYO5A*, and *RAB27A*, we discovered a nonsynonymous mutation within *MLPH*, which appeared to match the reported autosomal recessive nature of the lilac dilution. This mutation (NC_019458.2:g.3451931C>A) results in a premature stop codon being introduced early in the protein (NP_001139743.1:p.Glu14*), likely losing its function. Validation testing of additional lilac Jacob sheep and known carriers, unrelated to the original case, showed a complete concordance between the mutation and the dilution. This stop-gain mutation is likely the causative mutation for dilution within Jacob sheep.

## 1. Introduction

Coat color is suspected to be one of the first traits selected for in livestock species after domestication. Historically, selection in sheep has been for white wool, due to its ability to be dyed, as opposed to nonwhite wool. While white wool remains the dominant product in the commercial wool market, nonwhite wool can bring higher prices in niche markets. Nonwhite wool comes in a variety of patterns and colors, which breeders can select to increase the revenue from wool sales. One such nonwhite coat color variation is dilution, which is commonly represented by lighter shades of color pigmentation. A dilute phenotype has been observed within the Jacob breed, often called lilac. This dilution results in the nonwhite portions of the wool appearing gray, rather than the traditional black. Based on pedigree analysis of the Jacob Sheep Breeders Association registry, the dilution is inherited and expressed in an autosomal recessive pattern [1].

Dilute coat color phenotypes are commonly the result of impaired melanosome transport, leading to an irregular clustering of pigment. This irregular clustering of melanosomes results in decreased light absorption in the fiber, resulting in black hair or wool that appears grey. Melanophilin, together with myosin Va and Rab27a, form a protein complex that is responsible for transporting melanosomes to the cytoskeleton of melanocytes [2]. This complex has been shown to be required for proper melanosome transport [3]. Defects in any of these three genes, melanophilin (*MLPH*), myosin Va (*MYO5A*), and Rab27a (*RAB27A*), have been linked with several dilute phenotypes and the autosomal recessive Griscelli syndromes in humans (OMIM #214450, 607624, 609227) [4,5]. Of the Griscelli syndromes, type 3 (OMIM #609227) is linked to mutations within *MLPH* and is the only one of the three types to exhibit hypopigmentation in the absence of neurological or immunological abnormalities [5].

Until now, no dilute phenotypes in sheep or goats have been associated with any specific molecular variants. Dilute phenotypes observed in chickens (OMIA #001445-9031) [6], dogs (OMIA #000031-9615) [7,8,9], rabbits (OMIA #000031-9986) [10,11,12], cats (OMIA #000031-9685) [13], American minks (000031-452646) [14,15], and Belgian Blue cattle (OMIA #000031-9913) [16] have all been linked to variants within the melanophilin gene (*MLPH*). Therefore, *MLPH* was the most promising candidate gene to investigate for variants contributing to the dilute phenotype in Jacob sheep. The purpose of this study was to identify the genomic variant(s) responsible for the lilac color seen in Jacob sheep, using whole-genome sequencing and a candidate gene approach. By utilizing whole-genome sequencing of a known dilute case and a known carrier, a premature stop-gain point mutation was identified within *MLPH*. This work adds to our knowledge of *MLPH* mutations, leading to dilute phenotypes in domestic species.

## 2. Materials and Methods 

### 2.1. Sample Collection

All sheep were sampled in accordance with the Cornell University Institutional Animal Care & Use Committee (Protocol #2014-0121). Owner consent was given prior to sample collection for privately owned sheep. Whole blood was collected from the jugular vein via 10-mL vacutainers containing K_2_EDTA, and genomic DNA was extracted following the Qiagen Puregene Protocol (Gentra Systems, Inc. Minneapolis, MN, USA). The genomic DNA was stored at −80 °C until it was sequenced.

The dilute phenotype was visually characterized by diluted pigment in the nonwhite portions of the fleece. An example of dilute and non-dilute Jacobs can be seen in Figure 1. The two Jacobs used for whole-genome sequencing were unrelated within five generations. Additional dilute Jacobs were sourced from an unrelated flock. In total, 22 dilute Jacobs, 13 known carriers, and 26 non-dilute Jacobs were available for testing. The carriers were determined via pedigree analysis, using the Jacob Sheep Breeders Association pedigree database [17]. An additional 163 sheep that were not dilute, representing the Icelandic, Karakul, California Red, Romeldale, Romney, Finnsheep, Lincoln, and Shetland breeds, were used for validation testing.

### 2.2. Whole Genome Sequencing

TruSeq PCR-Free libraries were prepared for a known dilute case, a known dilute carrier, and sixteen additional non-dilute control samples from other breeds. The libraries were sequenced using 150 bp paired-end reads on an Illumina HiSeq X Ten platform to generate approximately 20x coverage per individual. These sequences have been deposited in NCBI’s sequence read archive (SRA) and can be found under the BioProject accession: PRJNA480684. Reads were aligned to Oar_v4.0 using the Burrows–Wheeler aligner mem algorithm, version 0.7.12-r1039 [18]. Average genome coverage was calculated using goleft covstats [19]. The alignments were locally realigned, and variants were called and filtered following the Genome Analysis Toolkit’s best practices, version 4.0.3.0 [20]. Small nucleotide variants were called using the HaploytpeCaller, within the Genome Analysis Toolkit. Variants were filtered from the analysis using the Genome Analysis Toolkit, with the following thresholds: quality depth < 2.0, Phred-scaled strand bias *p*-value > 60.0, mapping quality < 40.0, MQRankSum < −12.5, and ReadPosRankSum < −8.0. We also evaluated the predicted impact of these variants using SNPEff and the NCBI *Ovis aries* annotation release 102 [21]. Based on previous work in other domestic species [6,7,8,9,10,11,12,13,14,15,16], we focused our investigation on variants within *MLPH*. Variants within and surrounding *MLPH* (Oar_v4.0: OAR1:3,383,028-3,478,858) were filtered to be homozygous alternate in the dilute case, heterozygous in the known carrier, and homozygous reference in the sixteen non-dilute non-Jacob samples. Due to the nature of the melanosome transport and the known influence of *MYO5A* and *RAB27A*, we performed the same approach on the identified variants within *MYO5A* and *RAB27A*.

### 2.3. Candidate Variant Validation 

To validate the candidate variant, a PCR was designed to take advantage of the RFLP removed by the NC_019458.2:g.3451931C>A mutation. The following forward and reverse primers were designed to capture the first exon of *MLPH*, using Primer3 software [22]: F: 5′-GTCCCGCCACACACACTTAC-3′; R: 5′-TCGGTGTTTTCTGCATTGTC-3′. PCR amplification was performed in a 20 µL volume, containing 2 µL of DNA (diluted to a concentration of 25 ng/µL), 2 µL of 10× PCR reaction buffer with 20 mM MgCl2 (Roche Diagnostics), 0.2 µL of Taq DNA Polymerase [23], 2 µL of 2 mM dNTPs, 2 µL of forward and reverse 5 µM primers, and 9.8 µL PCR-grade water. The PCR was carried out in a BioRad T100 thermal cycler (BioRad Laboratories), with the following conditions: 3 min at 95 °C followed by 40 cycles of 30 s at 95 °C, 30 s at 59 °C, and 30 s at 72 °C, and a final extension time of 3 min. The PCR product was Sanger sequenced to validate the candidate variant, using one case, one carrier, and one wild-type individual. To genotype additional animals, the restriction digest used 10 µL of the 245 bp PCR product, 0.1 µL of *EarI* restriction enzyme [1.0 U per reaction, New England Biolabs (NEB), Ipswich, MA], 1 µL of 10× NEB CutSmart buffer, and 8.9 µL MilliQ of water to bring the reaction volume to 20 µL and was incubated for 16 h at 37 °C. The resulting products were visualized by agarose gel electrophoresis, using a 3% agarose gel, run for 40 min at 150 V, and a 100-bp standard ladder (New England BioLabs) as a reference. The normal allele (C) resulted in four expected fragments of 121, 57, 38, and 29 bp, while the dilute allele (A) resulted in only three expected fragments of 178, 38, and 29 bp.

## 3. Results

### 3.1. Whole-Genome Sequencing

Whole-genome sequencing generated 220,771,257 raw paired reads for the dilute case, 205,778,310 raw reads for the known carrier, and an average of 230,685,048 raw reads for the remaining sixteen individuals. After aligning the reads to the Oar_v4.0 reference genome, the dilute individual had an average genome coverage of 17.34×, the carrier 15.88×, while the remaining sixteen samples averaged 20.41×.

We identified 2572 small nucleotide variants within and surrounding *MLPH* (NC_019458.2:g.3383028-3478858). After 254 SNPs and 3 indels did not pass our variant filtering criteria, we were left with 2315 variants. Of these, only 37 SNPs and 4 indels were predicted to have an impact on the resulting *MLPH* protein. Within the SNPs, 15 were predicted to have a low impact, 21 a moderate impact, and only 1 was predicted to have a high impact. Of the indels, two were classified as low impact, and two were classified as high impact. The two high impact indels appear in the alternate state in every individual, which we suspect is the result of a reference assembly error, encompassing the homopolymer that these indels are located near. After filtering the impactful variants for variants that were homozygous alternate in the dilute animal, heterozygous in the known carrier, and homozygous reference in the sixteen other individuals, only the single nonsynonymous high impact SNP remained (NC_019458.2:g.3451931C>A). This SNP is located within exon 1, and results in a premature stop codon early in the protein (NP_001139743.1: p.Glu14*).

We identified 2994 small nucleotide variants within *MYO5A*, and 1173 small nucleotide variants within *RAB27A*. However, none of the identified variants exhibited the pattern of being a homozygous alternate in the dilute sheep, heterozygous in the known carriers, and a homozygous reference in the sixteen other individuals. They were not investigated further in relation to the dilution phenotype. A full list of variants discovered within these three genes can be found in Appendix A.

### 3.2. RFLP Validation

Using the RFLP genotyping test validation of the NC_019458.2:g.3451931C>A mutation within unrelated lilac cases and additional Jacob and non-Jacob non-dilute sheep, we found a complete concordance of the phenotype with the dilute genotype. See Figure 2 for the band sizes seen across the different genotypes. See Table 1 for *MLPH* NC_019458.2:g.3451931C>A genotype counts.

## 4. Discussion

Prior studies in dogs [7], cats [13], rabbits [12], and other domestic species have identified the mutations within *MLPH* as responsible for dilute coat colors. While dilution can impact both eumelanin and phaeomelanin, there have been no reports of dilute phaeomelanin phenotypes within sheep. The present study has likely identified the causative mutation for dilute coat color within the Jacob breed. This is the first report linking mutations within *MLPH* to a coat color phenotype within sheep. This adds to the limited knowledge in sheep coat color molecular genetics, as recent reports on causal variants have been limited to the Agouti [24], Extension [25,26], and Brown Loci [27,28].

The mechanism of dilution in sheep appears to be similar to mutations previously described as responsible for dilution in several species. In cats, the dilution is the result of a premature stop codon caused by a single base-pair deletion within the second exon of *MLPH* [13]. Within Belgian Blue cattle, a 10-bp deletion in the first *MLPH* exon results in a premature stop codon being introduced [16], differing from the point mutation described in the present study but still causing a loss-of-function mutation early in the protein. Exon-skipping within rabbits, caused by a frameshift mutation, also introduces a premature stop codon, which leads to a dilute phenotype [10]. These reported mutations, occurring early in the protein, fall within the R27BD domain. This is necessary for targeting *MLPH* to RAB27A, without which melanosomes are unable to be transported along actin filaments [29]. This is likely to result in the irregular clustering of melanosomes that cause the change in color.

Despite the limited molecular knowledge of coat color regulation in sheep, there are various shades of black and brown observed within other breeds, such as the Shetland, Icelandic, and Romeldale. Further study is needed to identify whether other mutations in *MLPH*, *MYO5A*, or *RAB27A* exist and contribute to the various shades seen in these breeds. Future work should investigate multiple species protein alignments, to extensively characterize the protein changes required for dilute coat color across species. The discovered variant within *MLPH* will be useful for Jacob breeders wishing to increase the frequency of the lilac color within their flocks through marker-assisted selection.

In conclusion, we have identified a stop-gain mutation (NC_019458.2:g.3451931C>A) within *MLPH*, which appears to be the causative mutation for the coat color dilution, called lilac, within Jacob sheep.

## Figures and Tables

**Figure 1 genes-11-00618-f001:**
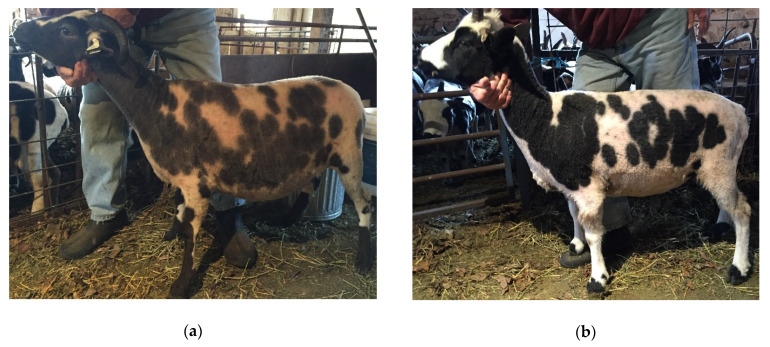
Photos of dilute (lilac) and non-dilute Jacob sheep that were used for whole-genome sequencing are in panels (**a**) and (**b**): (**a**) and (**c**) are examples of the lilac dilution in Jacob sheep; (**b**) and (**d**) are examples of non-dilute Jacob sheep.

**Figure 2 genes-11-00618-f002:**
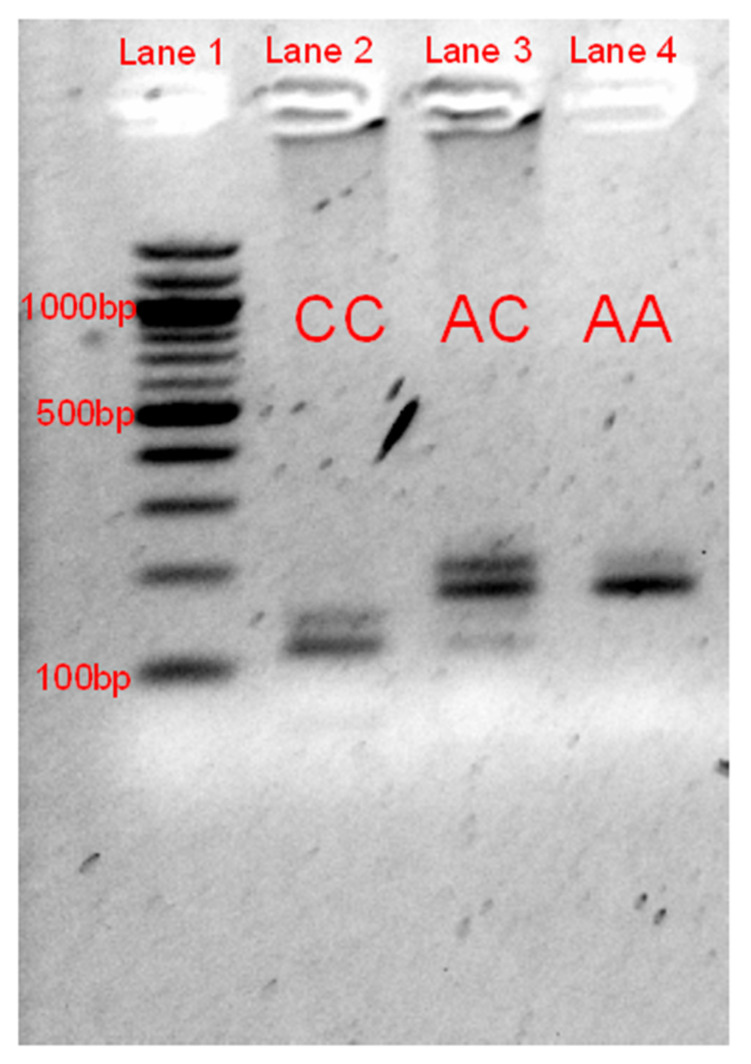
Gel image of the RFLP gel electrophoresis validation of the NC_019458.2:g.3451931C>A mutation. Lane 1 represents the 100-bp reference ladder, with the 1000, 500, and 100 bp bands labeled. Lanes 2 through 4 represent the three genotypes at this variant. The normal allele (C) results in four expected fragments of 121, 57, 38, and 29 bp while the dilute allele (A) results in only three expected fragments of 178, 38, and 29 bp. The 178- and 121-bp fragments were the primary ones used for genotyping, as the smaller bands were more difficult to identify. The bands of approximately 150 bp and 200 bp are likely to be the result of an incomplete digestion of the last *EarI* recognition site within the PCR product.

**Table 1 genes-11-00618-t001:** *MLPH* (NC_019458.2:g.3451931C>A) genotypes by dilute status.

Breed	Dilute Status	N	NC_019458.2:g.3451931C>A Genotype		
			AA	AC	CC
Jacob	Non-dilute	39	0	13	26
	Dilute	22	22	0	0
Non-Jacob	Non-dilute	163	0	0	163
Total		224	22	13	189

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
