# Peer review of "A Stop-Gain Mutation within MLPH Is Responsible for the Lilac Dilution Observed in Jacob Sheep"

_genes, 2020, doi:10.3390/genes11060618_

Round 1

Reviewer 1 Report

Dear Authors,

The aim of this paper was to identify the underlying genetic cause of a dilute coat colour variation in Jacob sheep. The paper undertook whole genome sequencing (WGS) of one dilute animal and an unrelated known carrier animal. After filtering variants based on genes of interest from this data and subsequent screening of sixteen additional control animals from different sheep breeds, a single variant within the Melanophilin (MLPH) gene was identified to segregate in dilute (lilac) sheep only.

This paper provided evidence of the first likely disease-causing variant within the MLPH gene that provides support for a dilute coat colour phenotype in sheep. The paper described the variant and made use of the local population to validate the variant, despite the small sample size.

Firstly, the Oar_v4.0_OAR1:3,451,931C>A nomenclature needs to be amended. Please refer to the HGVS nomenclature guidelines for best practice in reporting sequence variants both at the DNA, RNA and protein level. Transcript and protein IDs should also be included as per the guidelines.

The first paragraph of the introduction of the paper would benefit from some re-writing so it cohesively blends with the remaining paragraphs within the introduction. The materials and methods require more detailed descriptions. Within the methodology, additional information about the filtering parameters and software used for the WGS data is needed. No methodology was provided for the development of the PCR and RFLP genotyping assays and the ensuing screening of the C>A variant. This information must be provided to enable repeatability and should include solution concentrations of the materials used, cycling parameters etc.

It is not clear as to why other genes of interest apart from MLPH, MYO5A and RAB27A were not investigated. It is only stated that based on similar phenotypes in other species, these genes were prioritised. Whilst this makes sense to target genes of interest from similar phenotypic conditions in other species, the authors have not commented on other alternative routes to identify a more robust candidate gene list. For example, utilising a SNP genotyping chip and using a homozygosity mapping approach could identify runs of homozygosity common to dilute (lilac) sheep, which could then be investigated further through the analysis of whole genome sequencing variants within candidate genes. It should be noted as well, that the WGS data under accession PRJNA480684 could not be located. The authors do not state why only one dilute animal was sequenced, and why a parent of this animal was not sequenced, but instead, an unrelated known carrier animal was. The authors do not state how they know this animal is a known carrier and this needs to be clarified.

The results section of the paper would benefit from the inclusion of a table of variants identified in each candidate gene undergoing segregation with a recessive mode of inheritance, as no further detail is provided within the text apart from the number and type of variants identified from the WGS data. Further work involving in silico protein modelling or at the very least a multiple species alignment of the MLPH protein at the site of the variant would provide further support to the identified variant as disease-causing. This would allow for a more robust discussion and conclusion.

Please see below for more detailed comments:

Lines 16, 17, 110, 111, 118, 120 and 149:

Nomenclature should follow the HGVS guidelines.

Page 1, lines 25-27:

Consider re-wording, these sentences do not flow well. Whilst I understand the authors are referring to sheep, it would be useful to include this minor point in the text from the beginning. References are also needed to support the statements made in these sentences.

Page 1, line 39:

The full names of the genes should be stated first, followed by the gene abbreviation italicised within parentheses.

Page 1, line 40:

It would be useful to include the OMIM ID here for Griscelli syndrome in humans.

Page 2, lines 43 and 44:

It would be useful to include the OMIA IDs here for each species where available.

Figure 1:

Is a brighter image available for the dilute (lilac) sheep? The second image of the wildtype animal is very bright in comparison, and only slight changes in coat colour are visible between the dilute and wildtype animals.

Page 2, line 61:

How were the known carriers identified? Was this through pedigree analysis?

Page 2, line 62:

What additional sheep breeds were used for validation testing?

Page 3, line 81:

Provide references for these previous studies.

Page 3, line 83:

How were these filtered based on inheritance mode? Please list software.

Page 3, line 87:

Materials and methodology need to be included for both the PCR and RFLP performed and the instruments used. How many animals were used for this validation step before screening a wider subset of samples? Was Sanger sequencing conducted to confirm the PCR and RFLP targeted the variant of interest in affected, carrier and wildtype animals?

Page 3, line 91:

Provide parameters for gel electrophoresis.

Page 3, lines 92 and 93:

These lines should be moved to the results section. Is there any data available (i.e. gel images) to support these results?

Page 3, line 02:

A table of variants identified within MLPH, MYO5A and RAB27A should be included to enable readers to view these results.

Overall, the paper contributes knowledge to the field of coat colour genetics in identifying a likely disease-causing variant within dilute (lilac) sheep. This paper would significantly benefit from including additional information in relation to the materials and methods and results. Further work involving in silico protein modelling or at least multiple species alignment of the MLPH protein at the variant site will add further support. Furthermore, Sanger sequencing of the PCR and RFLP product would enable gold standard certainty regarding the robustness of these genotyping assays.

Reviewer 2 Report

The manuscript reports a nonsense mutation within the MLPH gene probably causal for coat color dilution in Jacob sheep. The mutation was detected by whole genome sequencing of one dilute, one carrier and 16 non dilute sheep and validated in a larger sample by RFLP analysis.

The manuscript is very straight forward and the results add to the knowledge of the genetic backgrounds of coat color dilution and enable a genetic test valuable for breeders. I have only some minor comments.

Minor revision

15 – “which exhibited the characteristics of an autosomal recessive inherited trait” This sentence is not optimal, please rephrase it.

42 – It should be “… in the absence of neurological or immunological symptoms”. In addition to coat color dilution, MYO5A mutations go along with neurological symptoms while RAB27A mutations cause immunological abnormalities.

45 – I would suggest to move the sentence starting with “However…” to the beginning of the paragraph and replace the “However” by “Until now” (or similar).

52 – It might be advantageous to add subheadings to structure this section.

104 – Please provide at least the information on all 37 SNPs and four indels with impact on MLPH protein detected in your study (location, impact on amino acid sequence, effect) as supplementary table. This might be valuable for further studies on (sheep) MLPH.

112 – Did you detect variants within MYO5A or RAB27A with impact on the protein? Then please provide a supplementary table with this information, too.

123 – There were two further variants (indels) with predicted high impact on the MLPH protein. It should be shortly discussed why they are supposed to have no causal effect on coat color (in addition to the fact, that only E14X segregates perfectly with the lilac color). I assume these indels were in the same phase as the causal variant and thus can be regarded to have occurred secondary. If they, however, should have been detected in control sheep, this would be an interesting hint on mutations with impact on MLPH protein but not causing color dilution.

Round 2

Reviewer 1 Report

Dear Authors,

Thank you for your detailed response and consideration to the previous requested amendments.

In regards to future work and support for the NC_019458.2:g.3451931C>A variant, if time does not permit to conduct a multiple species alignment (there are many software packages available to do this), I would suggest adding a sentence or two addressing this as future work within the concluding remarks of the paper.

A few minor editing comments:

Lines 41 -42:

There is no need to italicise the full gene name, only the abbreviation. For example: Melanophilin (MLPH). Please amend this for the remaining genes listed.

Line 48:

Please add the OMIA ID for chickens (OMIA 001445-9031).

Line 240:

Please remove the hyperlink from the reference.

Author Response

In regards to future work and support for the NC_019458.2:g.3451931C>A variant, if time does not permit to conduct a multiple species alignment (there are many software packages available to do this), I would suggest adding a sentence or two addressing this as future work within the concluding remarks of the paper.

Response: We have added a sentence on L187. "Future work should investigate multiple species protein alignments to extensively characterize the protein changes required for dilute coat color across species."

Lines 41 -42:

There is no need to italicise the full gene name, only the abbreviation. For example: Melanophilin (MLPH). Please amend this for the remaining genes listed.

Response: We have changed this in the text.

Line 48:

Please add the OMIA ID for chickens (OMIA 001445-9031).

Response: Thank you for providing the OMIA number. We have added this to the text.

Line 240:

Please remove the hyperlink from the reference.

Response: We have removed the hyperlink from the reference.